# Arbuscular Mycorrhizal Fungi and Glomalin Play a Crucial Role in Soil Aggregate Stability in Pb-Contaminated Soil

**DOI:** 10.3390/ijerph19095029

**Published:** 2022-04-21

**Authors:** Yinong Li, Jiazheng Xu, Jin Hu, Tianyu Zhang, Xuefeng Wu, Yurong Yang

**Affiliations:** 1Key Laboratory of Vegetation Ecology of the Ministry of Education, Institute of Grassland Science, Songnen Grassland Ecosystem National Observation and Research Station, Northeast Normal University, Changchun 130024, China; liyn129@nenu.edu.cn (Y.L.); xujz541@nenu.edu.cn (J.X.); huj791@nenu.edu.cn (J.H.); zhangty451@nenu.edu.cn (T.Z.); wuxf112@nenu.edu.cn (X.W.); 2State Environmental Protection Key Laboratory of Wetland Ecology and Vegetation Restoration, School of Environment, Northeast Normal University, Changchun 130117, China

**Keywords:** heavy metal, lead (Pb) pollution, symbiotic fungi, glomalin, soil aggregation

## Abstract

With the rapid development of industrialization and urbanization, soil contamination with heavy metal (HM) has gradually become a global environmental problem. Lead (Pb) is one of the most abundant toxic metals in soil and high concentrations of Pb can inhibit plant growth, harm human health, and damage soil properties, including quality and stability. Arbuscular mycorrhizal fungi (AMF) are a type of obligate symbiotic soil microorganism forming symbiotic associations with most terrestrial plants, which play an essential role in the remediation of HM-polluted soils. In this study, we investigated the effects of AMF on the stability of soil aggregates under Pb stress in a pot experiment. The results showed that the hyphal density (HLD) and spore density (SPD) of the AMF in the soil were significantly reduced at Pb stress levels of 1000 mg kg^−1^ and 2000 mg kg^−1^. AMF inoculation strongly improved the concentration of glomalin-related soil protein (GRSP). The percentage of soil particles >2 mm and 2–1 mm in the AMF-inoculation treatment was higher than that in the non-AMF-inoculation treatment, while the Pb stress increased the percentage of soil particles <0.053 mm and 0.25–0.53 mm. HLD, total glomalin-related soil protein (T-GRSP), and easily extractable glomalin-related soil protein (EE-GRSP) were the three dominant factors regulating the stability of the soil aggregates, based on the random forest model analysis. Furthermore, the structural equation modeling analysis indicated that the Pb stress exerted an indirect effect on the soil-aggregate stability by regulating the HLD or the GRSP, while only the GRSP had a direct effect on the mean weight diameter (MWD) and geometric mean diameter (GMD). The current study increases the understanding of the mechanism through which soil degradation is caused by Pb stress, and emphasizes the crucial importance of glomalin in maintaining the soil-aggregate stability in HM-contaminated ecosystems.

## 1. Introduction

Arbuscular mycorrhiza fungi (AMF) are a group of soil fungi with a wide distribution and important ecosystem functions. They can establish mutualistic symbiosis with approximately 80% of terrestrial plants [1], and they play an essential role in facilitating plant nutrient uptake, improving plant stress resistance, modifying soil structure, and restoring degraded ecosystems [2,3]. The extraradical hyphae formed by AMF can extend beyond the nutrient-deficient zone of the rhizosphere and enhance the absorption of phosphorus, nitrogen, and water. In turn, the host plant supplies 4–20% of the carbon hydration produced by photosynthesis to AMF to meet its growth and development needs [4]. In addition, AMF can also improve the adaptability of plants to environmental stresses by enhancing the expression of antioxidant enzymes, aquaporin, metallothionein, and other related genes [5,6,7]. AMF form a common mycorrhizal network among different plant species, thereby regulating important ecological processes such as nutrient distribution [8], plant competition [9], and community structure and succession [10]. However, the essential role of AMF in the stability of soil aggregates in degraded ecosystems remains unclear.

Soil is not only an important component of the terrestrial ecosystem, but also the source of the nutrients in our food supply. Soil structure is one of the most important soil properties, which determines structure, determines the retention, transformation, and transfer efficiency of water, air, heat, and nutrients in soil. The composition and stability of soil aggregates, which are the foundations of soil structure, are essential factors affecting soil fertility and ecosystem functions. Moreover, AMF hyphae and the produced glycoprotein (glomalin) can bind soil particles together through the “bonding–joining–packing” mechanism, and then act as bio-glue to promote the formation of large aggregates and increase the stability of the soil’s structure [11]. Glomalin is a special type of glycoprotein, produced specifically by hyphae and spores of AMF and released into the soil after decomposition; it is widely distributed, hydrophobic, insoluble, and recalcitrant in nature [12]. Previous studies indicated that AMF inoculation increased the GRSP content, mean weight diameter (MWD), and geometric mean diameter (GMD) of soil aggregates [13]. However, the relationships between the AMF growth index, GRSP, MWD, and GMD, and whether the effects of AMF on soil-aggregate stability are linked to GRSP, remain unclear.

Soil contamination by HMs constitutes a serious environmental problem. Pb is one of the most widely distributed and harmful toxic HMs in soil. China is the world’s largest producer of mineral Pb, with 63.9% of the world’s total mineral Pb production in 2020 (China Lead Industry Development Report 2020). In recent years, Pb contamination caused by mining, smelting, processing and other activities is increasing, and HM pollution incidents, such as blood Pb overload, have occurred from time to time [14]. In order to effectively address soil Pb contamination and restore degraded ecosystems, related research and practice have been comprehensively carried out in China. Most previous studies focused on distribution patterns [15,16], health-risk assessments [17,18], and remediation technology related to Pb-contaminated soil [19,20]. However, the positive effects of AMF on HM contamination as a bioremediation strategy to reduce the damage caused by Pb are still unclear. In view of this, a three-compartment root box was used in this study to investigate the effects of AMF inoculation on GRSP and the composition, and stability of soil aggregates under Pb stress. The objectives of this research are (1) to clarify the effects of Pb stress on AMF growth, GRSP, and the composition and stability of soil aggregates; and (2) to analyze of the effect of the AMF pathway on soil aggregates’ stability under Pb stress. This study provides a scientific basis for understanding the physiological and ecological functions of AMF and their potential value in restoring degraded ecosystems.

## 2. Materials and Methods

### 2.1. Preparation of Substrate Soil, Plant and AMF Inoculum

The substrate soil was prepared by collecting surface-layer (0–15 cm) soil from the experimental station of Jilin Agricultural University, Changchun, China (43°49′07″ N, 125°23′56″ E). The station has a continental monsoon climate with four distinctive seasons. The annual precipitation is about 600–700 mm and the annual average temperature is approximately 6.7 °C. The soil samples were passed through a 2-centimeter mesh sieve to remove any debris, rocks, and large organic matter, and were then air-dried at room temperature for 30 days. Subsequently, the soils were sterilized in clean cloth bags at 121 °C, 0.11 Mpa for 2 h (twice) to eliminate all microorganisms. A subsample was analyzed for physical and chemical properties of the soil. The soil belonged to the chernozem with pH 7.61, soil organic matter 25.7 mg g^−1^, total nitrogen 15.4 mg g^−1^, available nitrogen 93.6 mg kg^−1^, total phosphorus 0.86 mg g^−1^, available phosphorus 12.7 mg kg^−1^, and Pb 29.3 mg kg^−1^. The soils were equilibrated with Pb(NO_3_)_2_ solution at the final concentrations of 0, 500, 1000, and 2000 mg kg^−1^ Pb for 2 weeks, undergoing five cycles of saturation with deionized water and air-drying.

The seeds of *Bidens parviflora* Willd. [1,2] were collected from Longwan National Natural Reserve, Jilin Province, China (42°20′56″ N, 126°22′51″ E). They were surface-sterilized with 75% ethyl alcohol solutions for 10 min, and rinsed with distilled water five times. The seeds were placed on moistened filter paper in 9-centimeter-diameter Petri dishes. Four days later, the germinated seeds with uniform size were transplanted into plastic pots filled with 2.5 kg of autoclaved soil (15-centimeter upper diameter, 12-centimeter lower diameter, and 15 cm deep).

The AMF inoculum, *Funneliformis mosseae* (T.H. Nicolson & Gerd.) C. Walker & A. Schüßler (formerly Glomus mosseae) was obtained from Beijing Academy of Agriculture and Forestry Sciences (BAAFS). The fungus was propagated in pot culture with maize (*Zea mays* L.) and white clover plants grown in sterilized sand for 3 months before it was used for the experiment. At harvest, the aboveground parts of maize and white clover were discarded, and the dried substrate soils (36 spores g^−1^) were mixed homogeneously to use as the AMF inoculum to ensure that all treatments could receive the same number of spores. The AMF inoculum containing 500 spores weas placed 2 cm below the germinated seeds and covered with substrate soils. After emergence, seedlings were thinned to a final density of four plants per pot.

### 2.2. Experimental Design

The experiment was conducted in the greenhouse of Northeast Normal University, Changchun, Jilin Province, China (125°25′36″ E, 43°49′30″ N), and lasted 95 days. The experiment had a randomized complete block design with two AMF inoculation treatments (NM, non-mycorrhizal inoculation; AMF, AMF inoculation) and four Pb-addition treatments (Pb 0, 0 mg kg^−1^; Pb 500, 500 mg kg^−1^; Pb 1000, 1000 mg kg^−1^; Pb 2000, 2000 mg kg^−1^). Each treatment had six replicates for a total of 48 plastic pots. The pots were uniformly irrigated with distilled water every 2 days.

### 2.3. Sampling, Harvest, and Chemical Analysis

At harvest, plant root and soil samples in each pot were collected separately for further analysis.

Plant roots were harvested to stain using 0.05% trypan blue [21], and mycorrhizal colonization (MC) was assessed based on proportion of root length colonized by AMF under a microscope [22]. Hyphal length density (HLD) of AMF was determined by the modified grid-line intersection method described by Jakobsen et al. [23]. The soil sample (5 g) was blended with mixed distilled water (200 mL) and sodium hexametaphosphate in a beaker (500 mL). The hyphal suspension was then poured through 180-micrometer and 38-micrometer sieves. The filtrated materials were transferred to a beaker, shaken for 30 s, and then left to settle for 5 min at room temperature. The supernatant passed through a microporous membrane (0.45 μm) using vacuum pump. The hyphae were stained by a 0.05% (*w/v*) trypan blue solution, and measured according to the gridline intercept method at 200× magnification under a microscope. The HLD was expressed in units of m g^−1^ dry soil. 

The spore density (SPD) of AMF was determined by the wet sieving and decanting method proposed by Gerdemann and Nicolson [24]. Specifically, 50 g of soil samples was stirred evenly with 100 mL water, and allowed to settle for 30 min at room temperature. The suspension passed through a sequence of sieves (1000, 750, 500, and 38 μm). The procedure was repeated four times. The filtrated materials were transferred from the last two sieves to the 100-milliliter centrifuge tube with 60% sucrose solution, centrifuged at 3000 r/min for 5 min. The number of AMF spores was counted under a dissecting microscope and the spore density (SPD) was expressed as number of spores in 1 g of dry soil.

The concentrations of total glomalin-related soil protein (T-GRSP) and easily extractable glomalin-related soil protein (EE-GRSP) were determined according to procedures described by Wright and Upadhyaya [17]. The air-dried soil (1 g) was incubated with 8 mL of 20 mM sodium citrate solution (pH 7.0), autoclaved at 121 °C and 103 kPa for 30 min, and then centrifuged at 10,000× *g* for 15 min to extract EE-GRSP. By contrast, the T-GRSP was extracted with 8 mL of 50 mM sodium citrate solution (pH 8.0) by autoclaving at 121 °C and 103 kPa for 60 min. The procedure of extraction was repeated six times and all suspensions were collected. The EE-GRSP and T-GRSP concentrations were measured spectrophotometrically by the Bradford dye-binding assay using bovine serum albumin (BSA) as the standard.

Soil aggregates’ stability was determined by the wet sieve method [25]. The undisturbed soil samples were naturally air-dried at room temperature. A total of 50 g of soil samples was placed at the top of a stack of sieves (2 mm, 1 mm, 0.25 mm, and 0.053 mm) and soaked in a bucket overnight. The soils were sieved by raising and lowering at a distance of 5 cm and a stroke of 30 times min^−1^ for 10 min. Finally, the weights and the mass percentages of the soils with different soil-particle sizes were calculated. The mean weight diameter (MWD) and the geometric mean diameter (GMD) of the soils were used to evaluated the stability of the soil aggregates [26] and were calculated as follows:$$\mathrm{MWD}=\overset{n}{\underset{i=1}{\sum }}{\bar{x}}_{i}w_{i}$$
$$\mathrm{GMD}=\exp\left(\overset{n}{\underset{i=1}{\sum }}w_{i}\ln{\bar{x}}_{i}\right)$$
where w is the mass of each soil-particle size, $w_{i}$ is the mass percentage of each soil-particle size (%), and ${\bar{x}}_{i}$ is the average diameter of each soil-particle size.

### 2.4. Statistical Analysis

Prior to statistical analysis, the Kolmogorov–Smirnov test and the Levene test were applied to assess data normality and homogeneity, respectively, using SPSS 21.0. for Windows 10 (SPSS Inc., Chicago, IL, USA). Subsequently, one-way ANOVA was used to assess the significant differences in HLD, SPD, T-GRSP, EE-GRSP, mass percentage of soil particles, MWD, and GMD among different treatments, whereas two-way ANOVA was performed to reveal the interactive effects of Pb stress and AMF inoculation on T-GRSP, EE-GRSP, MWD, and GMD. Pearson’s correlation analysis was used to explore the strength of relationships among HLD, SPD, T-GRSP, EE-GRSP, mass percentage of soil particles, MWD, and GMD. The random forest model was performed to clarify the main factors affecting the stability of soil aggregates using the randomForest function from the randomForest package in R-4.1.1 version [27]. The pathways of Pb pollution affecting the stability of soil aggregates were revealed based on the structural equation model analysis using lavaan package in R-4.1.1 version [28].

## 3. Results

### 3.1. Effects of Pb Stress and AMF Inoculation on Plant Growth

The AMF inoculation positively affected the growth and development of the plants under the control and Pb-stress conditions (Figure 1). The shoot dry weight and root dry weight were higher in the mycorrhizal plants than in the non-mycorrhizal plants under all conditions, in spite of the fact that there were no differences in the R/S ratio. Additionally, the AMF inoculation significantly increased the plant height at the Pb 1000 stress level, while no difference was found at the other three Pb stress levels. These findings indicated that the AMF inoculation greatly improved the biomass rather than the height or R/S ratio of the *B. parviflora* seedlings under Pb-stress conditions.

### 3.2. Effects of Pb Stress on AMF Growth Parameters

The mycorrhizal colonization (MC), hyphal density (HLD), and spore density (SPD) of the AMF increased first, and then decreased with the increase in Pb stress levels (Figure 2). The maximum values of MC, HLD, and SPD were 64.07%, 0.81 m g^−1^ and 11.5 g^−1^, and all of them occurred at the Pb 500 level. The MC, HLD, and SPD at Pb 0 levels were significantly higher than at Pb 2000 levels. The Pearson’s correlation revealed that MC (r = −0.593; *p* = 0.002), HLD (r = −0.831; *p* < 0.001), and SPD (r = −0.609; *p* = 0.002) had considerable relationships with the Pb concentration.

### 3.3. Effects of Pb Stress and AMF Inoculation on Glomalin-Related Soil Protein Content

The AMF inoculation increased both the total glomalin-related soil protein (T-GRSP) and easily extractable glomalin-related soil protein (EE-GRSP) contents at all Pb stress levels, whereas the Pb stress exerted harmful effects on the production of glomalin-related soil protein (GRSP) by AMF (Figure 3). There was no significant difference in GRSP content under the control treatment. The correlation analysis showed that the T-GRSP (r = −0.921; *p* < 0.001) and EE-GRSP (r = −0.827; *p* < 0.001) in the AMF inoculation treatment were significantly and negatively correlated with the Pb concentration. The results of the two-way ANOVA indicated that the Pb stress and AMF inoculation had significantly interactive effects on the T-GRSP and EE-GRSP (*p* < 0.001).

### 3.4. Effect of Pb Stress and AMF Inoculation on Soil Aggregate Distribution

The percentage of soil aggregates 2~1 mm, 1~0.25 mm, and 0.25~0.053 mm decreased gradually with the increase in the Pb stress level, while no difference was found at Pb 0 and Pb 500 stress levels (Figure 4). By contrast, the percentage of <0.053 mm soil aggregates increased with the increase in Pb concentration, with a 99% increase at the Pb 2000 compared to the Pb 0 stress level. The change pattern of the percentage of soil aggregates under AMF inoculation treatment was similar to that under non-AMF inoculation treatment, and the percentage of soil aggregates >2 mm, 2~1 mm, and 1~0.25 mm decreased with the increase in Pb concentration. Moreover, the Pb stress enhanced the percentage of soil aggregates <0.053 mm, but had no effect on the percentage of soil aggregates 0.25~0.053 mm.

### 3.5. Effects of Pb Stress and AMF Inoculation on Soil Aggregate Stability

The Pb stress exerted a negative effect on the mean weight diameter (MWD) and geometric mean diameter (GMD), while these were higher under the AMF inoculation treatment compared with under the non-AMF inoculation treatment (Figure 5). The MWD and GMD decreased with increasing Pb concentrations under both the non-AMF and the AMF inoculation treatment. The MWD and GMD were significantly lower at Pb 1000 and Pb 2000 stress levels than at Pb 0 and Pb 500 stress levels. However, no difference in MWD and GMD was observed between Pb 0 and Pb 500 stress conditions. The correlation analysis showed that the MWD (*p* < 0.001) and GMD (*p* < 0.001) in both the non-AMF and the AMF inoculation treatment were significantly and negatively correlated with the Pb concentration. Individual Pb stress and AMF inoculation treatments had significant effects on the MWD and GMD, but their interactive effect was not significant on the soil-aggregate stability.

### 3.6. Correlations between AMF Status, Glomalin-Related Soil Protein, and Stability of Soil Aggregates

The results of the correlation analysis showed that there were significant correlations (*p* < 0.01) between the hyphal density (HLD), spore density (SPD), total glomalin-related soil protein (T-GRSP), easily extractable glomalin-related soil protein (EE-GRSP), mean weight diameter (MWD), and geometric mean diameter (GMD) (Figure 6, *p* < 0.001). To clarify the important roles of the AMF and GRSP in the stability of the soil aggregates, the random forest model was used to predict the soil-aggregate stability (Figure 7). The model qualified the significance test (*p* < 0.05) and explained 81.79% of the MWD and 74.28% of the GMD of soil aggregates. The HLD, EE-GRSP, and T-GRSP had significant effects on the stability of the soil aggregates, whereas the HLD and EE-GRSP explained the most important factors for soil aggregate stability, followed by the T-GRSP.

## 4. Discussion

### 4.1. Effect of Pb Stress on AMF Growth Parameters and Glomalin-Related Soil Proteins

Lead is an extremely dangerous and toxic pollutant for the growth and development of organisms, and it is widely distributed in nature. Pb stress induces the production of large amounts of reactive oxygen species (ROS) in biological cells, mainly including superoxide anion (O_2_^−^), hydrogen peroxide (H_2_O_2_), and hydroxyl radical (-OH). Small amounts of ROS generated by external stimulation during signal transmission can stimulate signaling pathways, participate in cellular signal transduction processes and activate antioxidant signaling pathways in the body [29]. However, the accumulation of ROS can lead to oxidative stress in cells, causing cell membrane degeneration, ion leakage, lipid peroxidation, DNA/RNA denaturation, and, eventually, cell lysis. In addition, Pb^2+^ can lead to reduced enzyme activity or even inactivation by both competing for the ion-binding site of the enzyme and inhibiting the active center of the enzyme, further aggravating the accumulation and toxicity of ROS in the cells. In this study, compared with the control treatment (0 mg kg^−1^ Pb), the hyphal density and spore density of AMF were significantly reduced at medium and high levels of Pb stress (1000 mg kg^−1^ and 2000 mg kg^−1^), but there was no significant difference at low levels of Pb stress (500 mg kg^−1^) (Figure 2). This was mainly due to the oxidative stress response of the AMF cells to medium and high concentrations of Pb stress, and the accumulation of ROS caused lipid peroxidation, cell membrane rupture, and an imbalance in the content and ratio of ions, which finally affected the growth and development of the AMF, resulting in a significant decrease in hyphal density and spore density. However, at low levels of Pb stress, Pb^2+^ induced an increase in the activities of the AMF cellular antioxidant substances (vitamins, glutathione, etc.) and antioxidant enzymes (oxide dismutase, ascorbate peroxidase, catalase, etc.), which eventually transformed the ROS into harmless H_2_O molecules through a series of chemical reactions, achieving a balance between ROS production and elimination. It can be seen that the AMF had a certain tolerance to Pb stress, which was closely related to the Pb concentration. When the Pb concentration exceeded 500 mg kg^−1^, the growth, development, and proliferation of the AMF were significantly inhibited. Additionally, the toxic effect of Pb largely depends on its bioavailability. The slightly alkaline condition (pH = 7.61) in this study might have caused the low mobility of the Pb in the soils, which could also partly explain the low toxic effects of the Pb on the AMF growth parameters [30,31,32].

Glomalin-related soil proteins (GRSP) are a special class of glycoprotein, specifically released by the hyphals and spores of AMF, which are abundant in soil and can be classified into two types: total glomalin-related soil proteins (T-GRSP) and easily extractable glomalin-related soil proteins (EE-GRSP) [12]. GRSPs have a long turnover time in soil and are not easily degradable. They play an important role in promoting soil organic carbon (TOC) accumulation, improving soil water and thermal conditions, improving the stability of soil aggregates, and regulating plant growth and community development. In this study, the T-GRSP and EE-GRSP contents gradually decreased with the increasing Pb stress, while there was no significant difference between the control treatment and the low-concentration Pb-stress treatment (Figure 3). These results were consistent with the pattern of the changes in the hyphal density and spore density of the AMF under Pb stress, indicating that the GRSP was a glycoprotein produced and secreted into the soil by the hyphals and spores of the AMF. A study by Yang et al. showed that both T-GRSP and EE-GRSP contents in AMF inoculation treatments were significantly and negatively correlated with heavy-metal Pb concentration, which was consistent with the results of this study [33]. Nevertheless, the present study found that there was no significant difference in the content of the GRSP (T-GRSP and EE-GRSP). This was mainly because the GRSP was specifically secreted by the AMF, and since there were no hyphals or spores in the uninoculated AMF treatment, no EE-GRSP was produced.

### 4.2. Effects of Pb Stress on Soil-Aggregate Stability

At present, studies on aggregates generally focus on the particle size, composition, and stability, nutrient content characteristics, and organic carbon content of aggregates, but the content and enrichment characteristics of heavy metals in aggregates and their effects on the stability of aggregates are rarely reported [34,35,36]. In this study, it was found that the Pb treatment significantly increased the proportion of soil grains <0.053 mm, while it significantly decreased the proportion of soil grains >2 mm and 2–1 mm, inhibiting the formation of soil macroaggregates (Figure 4), leading to a significant negative correlation between the Pb concentration and the soil-aggregate stability (*p* < 0.001). This was mainly due to the inhibitory effect of the Pb stress on the growth and development of the AMF (Figure 2), which significantly reduced the glomalin-related soil protein content released into the soil (Figure 3). The glomalin-related soil proteins, as special glycoproteins, played an important role in the formation of soil aggregates, which could bind soil particles together and then gradually form macroaggregate structures through the “bonding–joining–packing” hyphal mechanism, thereby improving the stability of the soil aggregates [37]. Therefore, the reduction in hyphal density and glomalin-related soil protein content caused by Pb stress might be the primary cause of decreases in soil aggregate stability.

### 4.3. Pathways of Pb Affecting Soil-Aggregate Stability

The colonization characteristics of AMF are important factors affecting the stability of soil aggregates. In order to reveal the relationship between AMF and the stability of soil aggregates under Pb stress, previous studies mostly used ANOVA and correlation analysis to analyze the direct relationship between the relevant factors, the mean weight diameter, and the geometric mean diameter while ignoring the complex interactions between these factors and failing to distinguish the possible direct or indirect pathways of action. In this study, we used random forest modeling and structural equation modeling to determine the mechanism through which Pb stress indirectly affects the stability of soil aggregates by impacting the AMF colonization characteristics. The results of the random forest model analysis showed that the hyphal density (HLD), easily extractable glomalin-related soil protein (EE-GRSP), total glomalin-related soil protein (T-GRSP) and spore density (SPD) had significant effects on the mean weight diameter (MWD) (*p* < 0.05), and the mean square error increases in the four characteristic variables were 6.47%, 6.42%, 5.40%, and 4.46%, respectively (Figure 7). Furthermore, only the HLD, EE-GRSP, and T-GRSP had a remarkable effect on the GMD (*p* < 0.05), with mean square error increases of 6.32%, 5.80%, and 5.29% for the three characteristic variables, respectively (Figure 7). These results indicated that the HLD, EE-GRSP, and T-GRSP were the dominant factors affecting the stability of the soil aggregates. These findings were consistent with previous findings that AMF colonization characteristics and specific secreted GRSP play an important role in soil-aggregate stability [38]. In this study, we further considered the interaction between multiple factors under Pb stress and used structural equation modeling to reveal the pathway through which Pb indirectly affects soil-aggregate stability through HLD and GRSP (Figure 8). The Pb stress had a negative direct effect on the HLD (−0.831) and GRSP (−0.679), and the GRSP had a positive direct effect on both the MWD (0.956) and the GMD (0.871), but the HLD had a positive indirect effect on the soil-aggregate stability through the GRSP (0.364). This might have been due to the fact that the Pb stress significantly inhibited the growth and development of the AMF, and the hyphal structure was disrupted, releasing a large amount of GRSP into the soil and weakening its direct contribution to the stability of the aggregates. Therefore, in HM-contaminated ecosystems, it is important to focus on maintaining the GRSP content in the soil, thus contributing to the improvement of the soil-aggregate stability and preventing the destruction of the soil structure.

## 5. Conclusions

Medium and high concentrations of Pb stress (>1000 mg kg^−1^) significantly inhibited the growth and development of AMF, with a significant decrease in MC, HLD, SPD, and AMF-secreted glomalin. Meanwhile, the AMF showed a certain tolerance to Pb stress, and there were no significant differences in the MC, HLD, SPD, and glomalin contents under low concentrations of Pb stress (500 mg kg^−1^) compared with the control. This was also possibly due to the low availability of Pb under slightly alkaline conditions. The Pb stress increased the mass percentages of fine sand, silt, and clay particles (<0.25 mm), while it decreased the mass percentages of gravel (>1 mm), resulting in a significant negative correlation between the soil-aggregate stability and the Pb concentration. AMF hyphals and glomalin play important roles in the formation of large soil aggregates, and HLD and glomalin content are both significantly and positively correlated with the stability of soil aggregates. According to the random forest model and structural equation model, we further determined that the HLD of the AMF, EE-GRSP, and T-GRSP were the most important factors driving the stability of the soil aggregates, while the Pb stress mainly affected the soil-aggregate stability indirectly, by regulating the HLD and the glomalin. The HLD also indirectly influenced the stability of the soil aggregates by regulating the glomalin. Therefore, focusing on the protection of glomalin and AMF is the key strategy to achieving rapid soil-structure restoration in degraded ecosystems polluted by HM.

## Figures and Tables

**Figure 1 ijerph-19-05029-f001:**
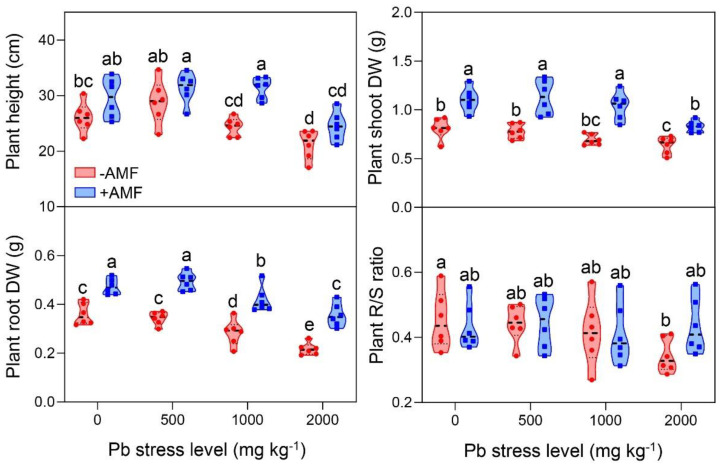
Plant height, shoot dry weight, root dry weight, and root/shoot (R/S) ratio of *B. parviflora* under different treatments. Dry weight (DW); Pb stress level (0, 500, 1000, and 2000 mg kg^−1^); non-mycorrhizal inoculation (−AMF); *F. mosseae* inoculation (+AMF). Data represent mean ± SD for biological replicates (*n* = 6). The same letter indicates no significant difference (Duncan’s test, *p* < 0.05).

**Figure 2 ijerph-19-05029-f002:**
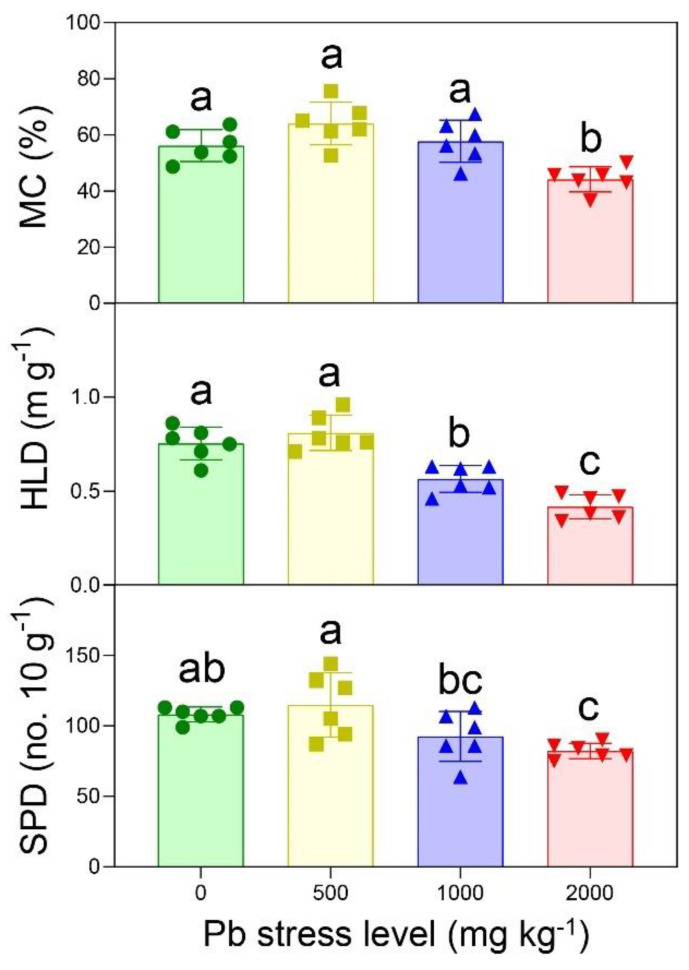
Mycorrhizal colonization (MC), spore density (SPD), and hyphal length density (HLD) of AMF under different Pb−stress levels (0, 500, 1000, and 2000 mg kg^−1^). Data represent mean ± SD for biological replicates (*n* = 6). The same letter indicates no significant difference (Duncan’s test, *p* < 0.05).

**Figure 3 ijerph-19-05029-f003:**
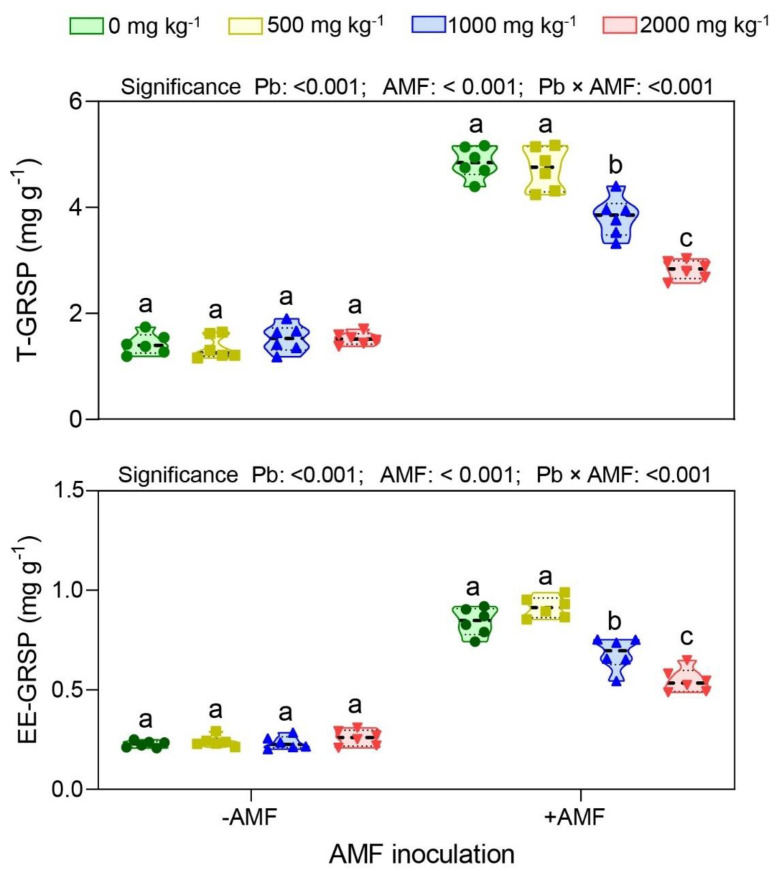
Total glomalin-related soil protein (T−-GRSP) and easily extractable glomalin-related soil protein (EE−GRSP) contents under different treatments. Non-AMF inoculation (−AMF); AMF inoculation (+AMF); Pb−stress level (0, 500, 1000, and 2000 mg kg^−1^). Data represent mean ± SD for biological replicates (*n* = 6). The same letter indicates no significant difference (Duncan’s test, *p* < 0.05).

**Figure 4 ijerph-19-05029-f004:**
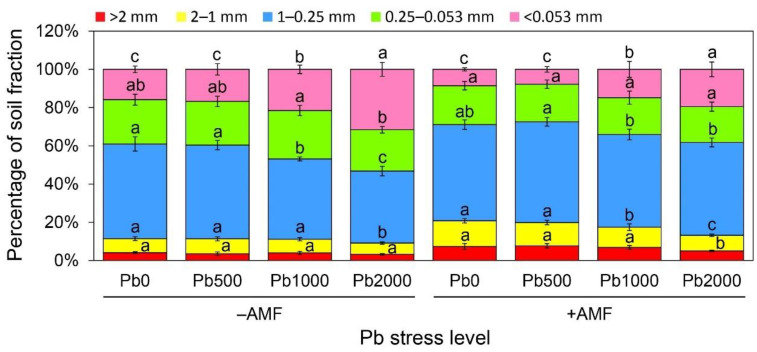
Soil-aggregate distribution under different treatments. Non-AMF inoculation (−AMF); AMF inoculation (+AMF); Pb stress level (0, 500, 1000, and 2000 mg kg^−1^). Data represent mean ± SD for biological replicates (*n* = 6). The same letter indicates no significant difference (Duncan’s test, *p* < 0.05).

**Figure 5 ijerph-19-05029-f005:**
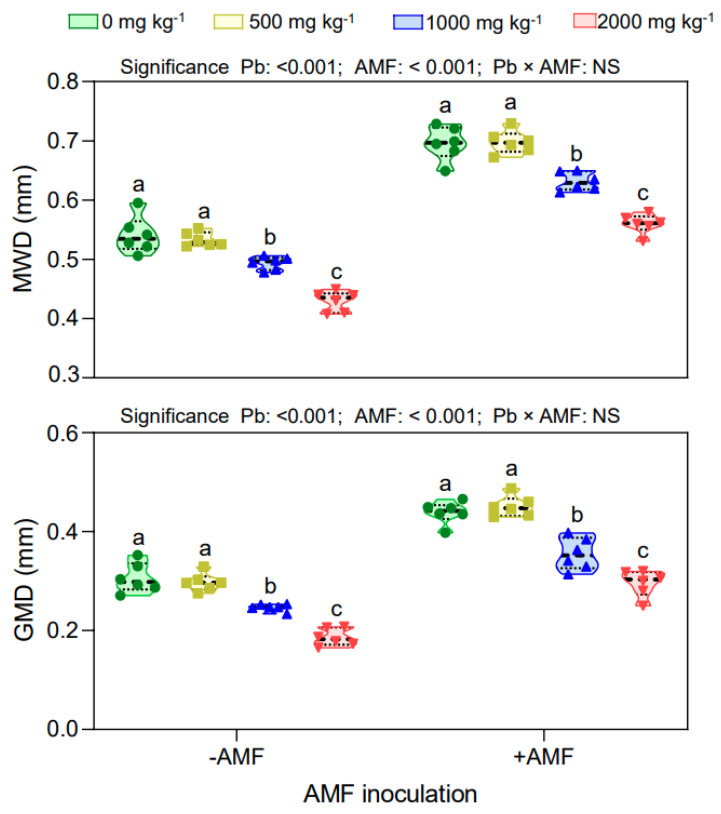
The mean weight diameter (MWD) and the geometric mean diameter (GMD) under different treatments. Non–AMF inoculation (−AMF); AMF inoculation (+AMF); Pb stress level (0, 500, 1000, and 2000 mg kg^−1^). Data represent mean ± SD for biological replicates (*n* = 6). The same letter indicates no significant difference (Duncan’s test, *p* < 0.05).

**Figure 6 ijerph-19-05029-f006:**
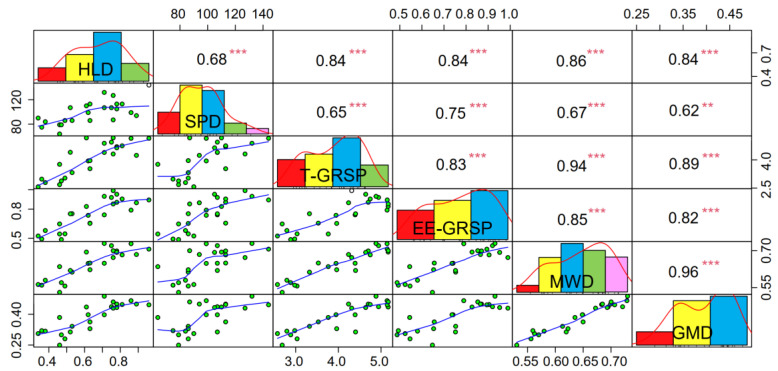
Correlation matrix among the selected variables related to AMF growth parameters, GRSP content, and soil-aggregate stability. HLD, hyphal length density; SPD, spore density; T-GRSP, total glomalin-related soil protein; EE-GRSP, easily extractable glomalin-related soil protein; MWD, mean weight diameter; GMD, geometric mean diameter (*** *p* < 0.001; ** *p* < 0.01).

**Figure 7 ijerph-19-05029-f007:**
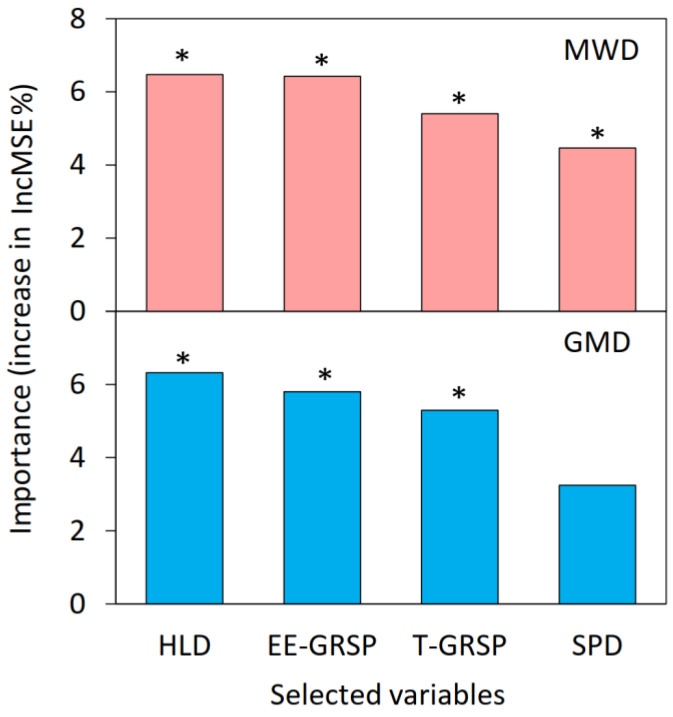
Random forest variable importance plot. The variables are ranked in order of relevance in predicting soil aggregate stability (MWD and GMD). The importance measure considered for the analysis is the mean decrease in accuracy computed via random forest classification algorithm. HLD, hyphal length density; SPD, spore density; T-GRSP, total glomalin-related soil protein; EE-GRSP, easily extractable glomalin-related soil protein (* *p* < 0.05).

**Figure 8 ijerph-19-05029-f008:**
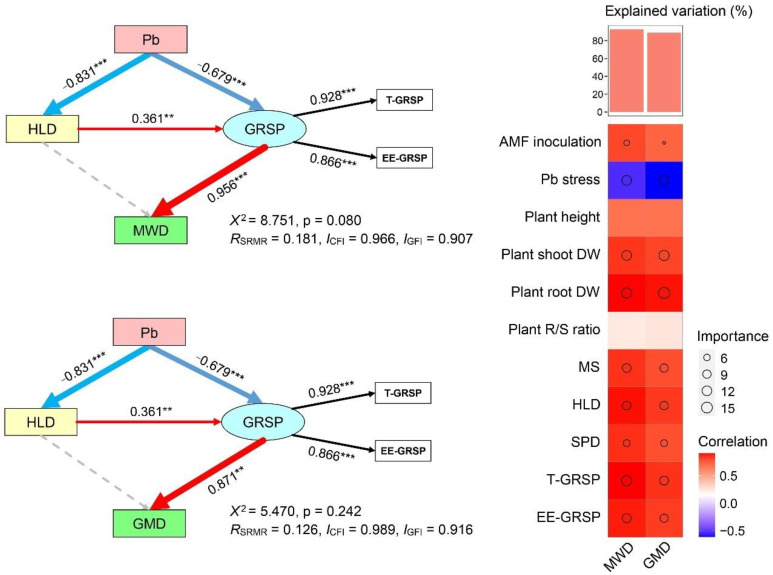
Structural equation model (SEM) illustrating the effects of Pb on soil-aggregate stability, and random forest variable importance score for all analyzed variables. Continuous and dashed arrows represent the significant and non-significant relationships, respectively. Adjacent numbers that are labeled in the same direction as the arrows represent path coefficients, and the width of the arrows are in proportion to the degree of path coefficients. Red and blue arrows indicate positive and negative relationships, respectively. Significance levels are denoted by ** *p* < 0.01 and *** *p* < 0.001.

## Data Availability

The datasets used and/or analyzed during the current study are available from the corresponding author on reasonable request.

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
