# Peer review of "Arbuscular Mycorrhizal Fungi and Glomalin Play a Crucial Role in Soil Aggregate Stability in Pb-Contaminated Soil"

_ijerph, 2022, doi:10.3390/ijerph19095029_

Round 1

Reviewer 1 Report

The work carried out in this document shows the role of the Arbuscular mycorrhizal fungi (AMF) in the presence of Pb. The results indicate that managing the amount of AMF greatly favors the size and diameter characteristics of the soil granules formed in the presence of Pb at different concentrations. This work is innovative as it shows the effect of the AMF inoculum on the degree of Pb assimilation, reducing in the same way the degree of toxicity of the heavy metal. In my opinion, this document is relevant, since there is not much information that describes the way in which the symbiosis of AMF stimulates the assimilation of Pb, improving the characteristics of the soils. I found that the article is relevant and presents a topic of scientific interest that has been little addressed. That's why I think it can be published.

Author Response

We appreciate your consideration of our manuscript “Arbuscular mycorrhizal fungi and glomalin play a crucial role in soil aggregate stability in Pb contaminated soil” (ijerph-1655374), and we appreciate the opportunity to address them in a revised version of the manuscript. Below we provide a detailed response to the comments provided by reviewers.

Point 1: The work carried out in this document shows the role of the Arbuscular mycorrhizal fungi (AMF) in the presence of Pb. The results indicate that managing the amount of AMF greatly favors the size and diameter characteristics of the soil granules formed in the presence of Pb at different concentrations. This work is innovative as it shows the effect of the AMF inoculum on the degree of Pb assimilation, reducing in the same way the degree of toxicity of the heavy metal. In my opinion, this document is relevant, since there is not much information that describes the way in which the symbiosis of AMF stimulates the assimilation of Pb, improving the characteristics of the soils. I found that the article is relevant and presents a topic of scientific interest that has been little addressed. That's why I think it can be published.

Response 1: Thank you for the time you have spent handling and reviewing our manuscript. We really appreciate your positive comments.

Reviewer 2 Report

Dear Authors,

Interesting work, making a significant contribution to the existing knowledge. Well written, but I have some minor comments:

 - in the description of the results, I would propose to explain the abbreviations, reading often you have to come back to the beginning to know what a given abbreviation means (it will not significantly lengthen the text, because it is short)

- in Figures 3 and 5: too much text and figures are unclear: transfer significance and Pb levels under pictures

- also the titles of the all Figures are not well understood: it can change e.g. non AMF inoculation (-AMF), AMF inoculation (+ AMF) and Pb stress level (0, 500, 1000, 2000 mg / kg)

- explain the abbreviations in the discussion, it will lengthen the discussion, but the text will be more understandable and easier to read.

Author Response

We appreciate your consideration of our manuscript “Arbuscular mycorrhizal fungi and glomalin play a crucial role in soil aggregate stability in Pb contaminated soil” (ijerph-1655374). All of the comments from the reviewers were very constructive and believe our revisions in response to these suggestions have greatly improved our manuscript. Point-by-point responses to each of the reviewers’ comments is listed below in this letter.

Point 1: In the description of the results, I would propose to explain the abbreviations, reading often you have to come back to the beginning to know what a given abbreviation means (it will not significantly lengthen the text, because it is short)

Response 1: The abbreviations have been explained throughout the Section of Results (see in Lines 185-186, Lines 212-216, Lines 248-249, Lines 266-271).

Point 2: In Figures 3 and 5: too much text and figures are unclear: transfer significance and Pb levels under pictures

Response 2: We have revised the text of significance levels as your suggestion in Figure 3 and 5.

Point 3: Also the titles of the all Figures are not well understood: it can change e.g. non AMF inoculation (-AMF), AMF inoculation (+ AMF) and Pb stress level (0, 500, 1000, 2000 mg / kg)

Response 3: We have revised the title of the figures as your suggestion to make them more readable.

Point 4: Explain the abbreviations in the discussion, it will lengthen the discussion, but the text will be more understandable and easier to read.

Response 4: The abbreviations have been explained throughout the Section of Discussion (Line 288, Line 301, Lines 307-308, Line 329, Lines 352-358, Lines 364-372).

Reviewer 3 Report

The presented manuscript focused on the role of arbuscular mycorrhizal (AM) fungi and glomalin in soil aggregate stability in Pb contaminated soils.

Heavy metals like Pb contamination of soil due to industrialization and other human activities has become an environmental problem with consequent problems for the human population. AM fungi can act as a filtration barrier against the transfer of heavy metals to plant shoots. The protection and enhanced capability of uptake of minerals result in greater biomass production.

Besides the effect of improving plant growth, AM fungi were also shown to play important roles in the formation and maintenance of soil aggregates. The extensive hyphal networks are involved in soil physical processes by crosslinking soil particles to macroaggregates. Glomalin, a glycoprotein at least partly produced by AM fungi, has been reported to play an important role in structuring soil. Several studies have demonstrated the positive correlation between the content of glomalin-related soil protein (GRSP) and soil aggregates.

The article submitted for evaluation is well-written and raises an interesting problem, however, the authors have not been avoided errors or understatements.

My main objection to the research conducted is the choice of soil for the experiment. The high pH (7.61) of the soil used causes a significant reduction in the mobility of lead in the soil. Up to 80% of the Pb contained in the soil may precipitate under these conditions and be unavailable to organisms, both fungi and plants. Therefore, the authors must take this fact into account when interpreting the results obtained. I suggest analysing the availability of Pb in soil using CaCl2 or EDTA.

LL201-202: It is not true that the level of MC was significantly higher at Pb0 and Pb500 compared to the other two levels (it is higher only with Pb2000), as is SPD, which level was statistically significantly different at Pb0 and Pb2000 and Pb500 and Pb1000 and 2000. The small differences between sequential Pb concentrations may be due to the low availability of this element.

Figure 4.: The shade of blue in the legend does not correspond to the blue in the graph, which can be confusing for the reader.

LL297-298: This finding may be more related to Pb availability than Pb tolerance by AMF.

L311: “Yang” not “yang”

LL314-315: According to the statistical analyses performed, the statement "the present study found that soil T-GRSP in the uninoculated AMF treatment increased with increasing Pb concentration" is not true (see Figure 3).

LL377-379: The observed lack of differences in MC, HLD, SPD and glomalin at Pb500 and control may be more due to the low bioavailability of this element than to AMF tolerance to Pb.

Author Response

We appreciate the comments from reviewers aimed at improving the quality of our manuscript entitled “Arbuscular mycorrhizal fungi and glomalin play a crucial role in soil aggregate stability in Pb contaminated soil” (ijerph-1655374), and we appreciate the opportunity to address them in a revised version of the manuscript. We revised the sections of Results and Discussion, which incorporate all suggestions from you and the two reviewers. The main conclusions remain the same and the mechanisms for AMF to reduce the damage of Pb are more strongly supported now. Below we provide a detailed response to the comments provided by the reviewers (text in red below), with explanations of the changes we made and their locations in the text.

Point 1: My main objection to the research conducted is the choice of soil for the experiment. The high pH (7.61) of the soil used causes a significant reduction in the mobility of lead in the soil. Up to 80% of the Pb contained in the soil may precipitate under these conditions and be unavailable to organisms, both fungi and plants. Therefore, the authors must take this fact into account when interpreting the results obtained. I suggest analysing the availability of Pb in soil using CaCl2 or EDTA.

Response 1: Thanks for this important point. We fully agree with you that that the bioaccessible concentration of Pb is more important than the total concentration. Therefore, we intended to conduct an additional experiment to measure the availability of Pb in soils, whereas, now it is impossible to do that because of pandemic (COVID-19). Currently, everyone is isolated at home or dormitory under the guidance of doctors and volunteers to minimize spread of the virus. Anyway, we have checked many studies that assessed the availability of Pb in relative high pH (7-7.9) and it is around 0 to 35.36%, with a mean of 7.7% [30-33]. Moreover, we have explained this important point in our revised paper to make it more readable (Lines 315-318).

Reference

  1. Finzgar N, Kos B, Lestan D. Bioavailability and mobility of Pb after soil treatment with different remediation methods. Plant Soil Environ. 2006, 1, 25-34.
  2. Wieczorek J.; Baran A.; Urban´ski K.; Mazurek R.; Klimowicz-Pawlas A. Assessment of the pollution and ecological risk of lead and cadmium in soils. Geochem. Hlth. 2018, 40, 2325-2342.
  3. Ma J.; Hao Z.; Sun Y.; Liu B.; Jing W.; Du J.; Li J. Heavy metal concentrations differ along wetland-to-grassland soils: a case study in an ecological transition zone in Hulunbuir, Inner Mongolia. Soil Sediment 2022, 1-12.

Point 2: LL201-202: It is not true that the level of MC was significantly higher at Pb0 and Pb500 compared to the other two levels (it is higher only with Pb2000), as is SPD, which level was statistically significantly different at Pb0 and Pb2000 and Pb500 and Pb1000 and 2000. The small differences between sequential Pb concentrations may be due to the low availability of this element.

Response 2: Thanks for this importane point. It has been rewriteen to make it more accurate (Lines 202-203).

Point 3: Figure 4.: The shade of blue in the legend does not correspond to the blue in the graph, which can be confusing for the reader.

Response 3: The color of the legend has been corrected in Figure 4.

Point 4: LL297-298: This finding may be more related to Pb availability than Pb tolerance by AMF.

Response 4: Thanks for this important point. We fully agree with you that that the bioaccessible concentration of Pb is more important than the total concentration. But now it is impossible to measure it because of pandemic (COVID-19) as we said in Point 1. Anyway, we have checked the published papers online and found that the Pb availability in relatively high pH is about 7.7%. Additionally, we have added this information into the revised paper to avoid possible misunderstanding (Lines 315-318).

Point 5: L311: “Yang” not “yang”

Response 5: “yang” has been changed to “Yang” in L331.

Point 6: LL314-315: According to the statistical analyses performed, the statement "the present study found that soil T-GRSP in the uninoculated AMF treatment increased with increasing Pb concentration" is not true (see Figure 3).

Response 6: We have changed "T-GRSP" to "GRSP". There was no significant difference in the content of GRSP and more explanation has been added into our revised paper (Lines 334-336).

Point 7: LL377-379: The observed lack of differences in MC, HLD, SPD and glomalin at Pb500 and control may be more due to the low bioavailability of this element than to AMF tolerance to Pb.

Response 7: Thanks a lot for your suggestion. We fully agree with you that that the bioaccessible concentration of Pb is more important than the total concentration. But now it is impossible to measure it because of pandemic (COVID-19) as we said in Point 1. Anyway, we have checked the published papers online and found that the Pb availability in relatively high pH is about 7.7%. Additionally, we have added this information into the revised paper to avoid possible misunderstanding. We have improved the section on conclusion and highlighted the effect of Pb availability on them (see lines 402-403). Thus, we feel our consideration of AMF to reduce the damage of Pb is more comprehensive.

Round 2

Reviewer 3 Report

I find a significant improvement in the quality of the text. Although, I found minor mistakes that need to be corrected. After the authors have made corrections, I recommend the manuscript for publication in the International Journal of Environmental Research and Public Health.

L20 and L307: Missing superscript on units

I suggest changing the keywords as the current ones overlap with the title.

Author Response

We appreciate the comments from reviewers aimed at improving the quality of our manuscript entitled “Arbuscular mycorrhizal fungi and glomalin play a crucial role in soil aggregate stability in Pb contaminated soil” (ijerph-1655374), and we really appreciate your positive comments. We revised the mistakes and changed the overlapped key works. Below we provide a detailed response to the comments provided by the reviewer (text in red below), with explanations of the changes we made and their locations in the text.

Point 1: L20 and L307: Missing superscript on units

Response 1: The superscript on units has been revised throughout the paper (Lines 20 and 314)

Point 2: I suggest changing the keywords as the current ones overlap with the title.

Response 2: Thanks for this important point. We changed key words “heavy metal”, “arbuscular mycorrhizal fungi”, “glomalin”, “soil structure” and “soil aggregate stability” to “heavy metal”, “lead (Pb) pollution” “symbiotic fungi”, “glomalin” and “soil aggregation” (Lines 32-33)